# Platelet-Rich Plasma Injections Decrease the Need for Any Surgical Procedure for Chronic Epicondylitis versus Conservative Treatment—A Comparative Study with Long-Term Follow-Up

**DOI:** 10.3390/jcm12010102

**Published:** 2022-12-23

**Authors:** Juho Aleksi Annaniemi, Jüri Pere, Salvatore Giordano

**Affiliations:** 1Department of Surgery, Welfare District of Forssa, 30100 Forssa, Finland; 2Department of Plastic and General Surgery, Turku University Hospital, University of Turku, 20500 Turku, Finland

**Keywords:** epicondylitis, platelet-rich plasma, injection therapy, nonoperative treatment, physical therapy, tennis elbow, lateral epicondylitis

## Abstract

Background: Platelet-rich plasma (PRP) injections may alleviate symptoms of chronic medial or lateral epicondylitis. Methods: We retrospectively analyzed a total of 55 patients with chronic ME or LE who had undergone at least 6 months of any conservative treatment before intervention. The patients were divided into two groups: the PRP group (*n* = 25), who received a single injection of autologous PRP to the medial or lateral epicondyle, and the PT group (*n* = 30), who continued with PT and pain medication. The primary outcome measures were pain and functional outcomes measured in terms of the following: Patient Related Tennis Elbow Evaluation (PRTEE), Visual Analogue Scale (VAS), and Disabilities of the Arm, Shoulder, and Hand (DASH), which were detected at preintervention, 6-, 12-, 24-, and 36-month follow-up. Secondary outcomes included complications and the need for any surgery at follow-up. Results: Primary outcome measurements showed significantly better results favoring the PRP group (6-month PRTEE total 43.2 ± 19.2 vs. 62.8 ± 24.0, *p* < 0.001; 12-month PRTEE total 6.9 ± 15.0 vs. 28.1 ± 24.4, *p* < 0.001; 24-month PRTEE total 4.8 ± 9.8 vs. 12.7 ± 14.5, *p* = 0.029), and significantly better results in VAS and DASH sub-scores. The PRP group required significantly fewer surgical procedures (*n* = 0/0% vs. *n* = 6/20%, *p* = 0.027) at follow-up (mean 38.3 ± 12.3 months), and one case of prolonged pain after injection was detected. Conclusions: Patients who underwent PRP injections for epicondylitis resulted in better pain and functional outcomes compared to physiotherapy, and this improvement lasted at least 24 months. They required fewer surgical procedures and achieved faster recovery than the PT group. We recommend PRP for chronic epicondylitis of the elbow before considering surgery when other treatments have failed.

## 1. Introduction

The etiology of medial epicondylitis (ME) or lateral epicondylitis (LE) of the elbow is not completely understood. It might be due to multiple factors from mechanical strain to abnormal microvascular response, which causes microtrauma in the common origin of the wrist and finger extensors (LE) or flexors (ME) [1]. Recent disease burden studies estimate that the prevalence in the general population of lateral epicondylitis is between 1–3%, and for medial epicondylitis, 0.4% [2,3,4].

Treatment options for LE and ME include non-steroidal anti-inflammatory drugs (NSAIDs), physical therapy, rest, corticosteroid injections, surgical intervention, extracorporeal shock wave therapy, proximal forearm strap, watchful waiting, and botulin toxin injections. [1,2,3,4]. Recently, platelet-rich plasma (PRP) has been widely studied for the treatment of epicondylitis. Results have been encouraging but also controversial, as there are both original randomized controlled trial (RCT) studies and meta-analyses that either approve or disapprove of the use of PRP in the treatment of elbow epicondylitis [5,6,7,8,9,10,11,12,13,14,15,16,17,18,19,20,21,22,23,24,25,26,27]. Most of the studies seem to favor PRP over other treatments [5,6,7,8,9,10,11,12,13,14,15,16]. Original studies against PRP are sparse, and some of them have limitations [22,23,24,25], mostly including patients with non-chronic epicondylitis. Studies specifically focused on chronic epicondylitis appeared to have more statistically significant results [9,10,11,12,13,14,15,16], and most of the meta-analyses seem to favor PRP over other treatment modalities [17,18,19,20,21].

The clinical course of LE is known to be self-limiting in most cases, but some patients develop chronic epicondylitis that may be clinically highly resilient to any previous conventional treatment options; this usually leads to more radical treatments such as surgery [4,26,27]. Currently, nonoperative treatment is the primary treatment modality, with surgery being reserved for patients who do not respond well enough to conservative treatments, because surgical outcomes have not resulted in optimal outcomes [4,26]. Patients who develop chronic epicondylitis suffer the most, poorly respond to NSAIDs, physical therapy, or other common treatment modalities. This is the patient group that requires the most help.

The key clinical questions are the following: Is PRP useful in chronic (medial and lateral) epicondylitis of the elbow? How long does the effect last? Will the patient with chronic epicondylitis avoid surgery with PRP treatment better than the patient with conventional treatments? Are there relapses during long-term follow-up? The current literature does not have data beyond two years of follow-up. There are some studies that compared PRP with surgery and corticosteroid injections; however, corticosteroids are not recommended anymore, and surgery is the last option. Instead, physical therapy is currently considered the primary mode of care [4,21,28]. There are no viable studies comparing PRP with physical therapy combined with NSAIDs, despite physical therapy being one of the cornerstones of nonoperative treatment.

In the present study, we aimed to analyze patients with chronic lateral or medial epicondylitis who had received physical therapy and pain medications for at least 6 months without improvements in symptoms. Then, these patients either received a single injection of PRP, or continued with physical therapy and NSAIDs. We hypothesized that patients treated with PRP would have their symptoms significantly reduced during the follow-up, and would avoid the need for surgery more often than patients in the conventional treatment group. Moreover, we also hypothesized that the effects of PRP would be beneficial in long-term follow-up.

## 2. Materials and Methods

This study was approved by the Institutional Review Board (39/13.01.01/2018, Welfare District of Forssa, Finland), and the ethical principles of the World Medical Association Declaration of Helsinki were followed.

We analyzed 55 consecutive patients with chronic LE or ME treated between 2014 and 2020 at the Department of Surgery, in the Welfare District of Forssa (public hospital), Finland. This was a single-center, retrospective study and, for purpose of this study, patients were divided into two groups: patients in the PRP group received a single injection of two milliliters of autologous PRP (Commercial Glo PRP kit, GloFinn corporation, Salo, Finland) to the medial or lateral epicondyle area of the elbow where respective extensor or flexor muscle insertions lay, according to the symptomatology, while the control group continued physical therapy (PT) and NSAID treatment.

Patients were referred for orthopedic consultation by primary healthcare, and they were given option to try physical therapy or PRP injection therapy to treat their condition. 

The PRP preparation protocol was as follows: 9 mL of venous blood was drawn from the patient, and a red blood cell (RBC) collector was connected to the syringe. The blood was centrifuged at 1200 revolutions per minute (rpm) for 5 min. Excess RBCs were discarded, and a second centrifugation was performed for 10 min at 1200 rpm. White blood cells were not separated from the PRP. The final product contained approximately 1–2 mL of PRP, with four to eight times higher platelet concentration than the normal physiological level. PRP injections were given at 2-week intervals.

The Injection was performed by an experienced orthopedist using anatomical landmarks without ultrasound guidance.

Physical therapy for the lateral/medial epicondylitis included wrist and finger extensor and flexor stretching, wrist rotations, handshake stretches, wrist curls, finger stretching (abduction), and gripping/squeezing a soft object (softball). The exercises were performed either with or without weight up to half a kilogram, depending on the individual’s level of strength. An experienced physical therapist instructed and motivated the patients before the intervention point and after that, patients were instructed to execute the exercises at least three times a day. There were also prescribed pain medications (ibuprofen 600 mg three times a day, and/or APAP 1 g maximum, three times a day).

Both groups underwent physical therapy, with NSAIDs (e.g., ibuprofen) and acetaminophen (paracetamol, APAP) for pain management, before considering any surgery as part of the typical conservative treatment. 

Indication for surgical intervention for elbow epicondylitis was at the surgeon’s discretion, when failure of prolonged nonoperative treatment (in this study, physical therapy and pain medication) persisted for more than 6 months. The surgical procedure consisted of open release and debridement of the extensor carpi radialis brevis, with decortication of the epicondyle. After surgery, the arm was splinted for 2 weeks, and normal mobilization was allowed with a return to maximum loading after 3–6 months.

The patients’ flowchart is shown in Figure 1. A total of 55 patients (PRP *n* = 25, PT *n* = 30) were included in the final analysis. The inclusion criteria were an age between 18 and 90 years; diagnosis with chronic ME or LE (symptom duration 6 months or longer) that did not respond to conservative treatment; and a preintervention Visual Analogue Scale (VAS) of 30–100. The exclusion criteria included cervical spine disorders or distal upper extremity neurological problems or conditions (e.g., carpal tunnel syndrome, ulnar nerve entrapment, neurological diseases affecting upper extremities, cervical spine radiculopathy); fractures or recent trauma of the upper extremities; pregnancy or possible pregnancy; previous other injection therapies of the elbow area (prior to 6 months); elbow joint arthrosis; major systemic disorders (e.g., hematological diseases, infections, immunodeficiency); previous surgery of the elbow (e.g., surgery due to epicondylitis or trauma); and patients without chronic epicondylitis. The patients enrolled in this study had not received enough benefits from physical therapy, had chronic lateral or medial epicondylitis, and, thus, either received PRP injections or continued with physical therapy. Patients’ demographics, treatment modalities, symptom duration, and clinical outcomes were collected from the patients’ electronic medical records.

The primary outcome measures were pain and functional outcomes, which were measured in terms of the following: Patient Related Tennis Elbow Evaluation (PRTEE), Visual Analogue Scale (VAS), and Disabilities of the Arm, Shoulder, and Hand (DASH). The patient data were collected before the intervention, at 6 months, 12 months, 24 months, 36 months, and at the final follow-up. Secondary outcomes included the need for surgical intervention during the follow-up, and complications. Continuous parametric and nonparametric data were reported as mean ± standard deviation (SD), while discrete data were reported using percentages. Statistically significance was set as whether the two-sided *p* value was ≤0.05 on a 95% confidence interval. Comparisons between the study groups were carried out using Student’s *t*-test for continuous variables and Fisher’s exact test for discrete variables, according to the nature of the data. All of the statistical analyses were conducted using SPSS statistical software (IBM SPSS Statistics, version 28, Armonk, NY, USA).

## 3. Results

Demographics of the patients are shown in Table 1. There was a significant difference in age (PRP 53.6 ± 8.4 vs. PT 48.4 ± 9.9, *p* = 0.045, Table 1) and sex ratio (PRP male:female ratio of 11:14 vs. PT male:female ratio of 23:7, *p* = 0.013). Most of the patients were affected by lateral epicondylitis (92% in the PRP group and 83% in the PT group, Table 1). Figure 2 describes the pre-interventional parameters, showing nonsignificant differences between the groups (Figure 2). The mean follow-up was 40.1 ± 6.5 months for the PRP group and 36.8 ± 15.5 months for the PT group, without significant differences between the groups (Figure 2).

There was a significant statistical difference favoring the PRP group at 6 months (in VAS, PRTEE, PRTEE function, PRTEE total, and DASH; Figure 3) and similarly, at 12 and 24 months (Figure 3). There were no significant differences between the groups at 24 months in PRTEE function (PRP 4.8 ± 9.0 vs. PT 11.1 ± 13.4, *p* = 0.061) and DASH (PRP 34.0 ± 8.0 vs. PT 39.9 ± 12.5, *p* = 0.052) (Figure 3). Likewise, there were no significant differences found in any of the scores at the 36-month follow-up (Figure 3). 

The PRP group had significantly fewer elbow surgical procedures than the PT group (*n* = 0/0% vs. *n* = 6/20%, *p* = 0.027) at follow-up (Figure 3). We detected one adverse effect in the PRP group, with one patient reporting prolonged pain of up to five days after the injection.

## 4. Discussion

The present study demonstrates that a single injection of PRP can significantly reduce the need for surgery, and improve pain and function scores in patients affected by elbow epicondylitis.

Previous studies on this topic suggest that PRP treatment is a viable option to avoid surgery for chronic medial or lateral epicondylitis [10,11,12,15]. The most consistent results supporting PRP treatments come from studies that focused on chronic epicondylitis [8,9,10,11,12,13,14,15,16]. Peerbooms et al. and Gosens et al. [5,6] reported the first large-scale RCT studies, with up to one- and two-year follow-ups of PRP, respectively, versus corticosteroid injections in chronic lateral epicondylitis; the outcomes favored PRP. Later, Mishra et al. [8] conducted a larger multicenter RTC study of chronic lateral epicondylitis patients, comparing PRP to active controls, and concluded that PRP is superior, suggesting that PRP should be the last line of treatment before considering any surgery. 

A few RCT studies compared using PRP versus autologous blood, saline injections, and/or corticosteroid injections to lateral epicondylitis, and showed no significant differences between the groups [24,25]. However, the study by Krogh et al. [25] reported a high dropout rate of patients in all of the study arms; hence, only data gathered from three months to the final analysis were included. Linnanmäki et al. reported the opposite results in a prospective randomized study without double blinding, and used leukocyte-poor PRP, which was also the most notable difference between their study and studies that favored PRP [5,6,8,24]. Fitzpatrick et al. [29] concluded in their meta-analysis that leukocyte-rich PRP is, thus, by far the best form of PRP to be used in the treatment of tendinopathies, and should be administered preferably as an intra-tendinous injection under ultrasound guidance. Surgery is usually not recommended, and is possibly even only harmful, as a double-blinded RCT study concluded [30].

Our outcomes indicated that the PRP group had statistically significant lower symptom scores than the PT group for the first two years of the follow-up, but not later. This may be due to growth factors in the PRP that may push the previously halted healing process forward. All the patients in this study suffered from chronic epicondylitis and had not received sufficient help from conventional treatments. Our results are consistent with the current literature, supporting the PRP treatment of chronic epicondylitis with PRP [8,9,10,11,12,13,14,15,16]. Interestingly, final outcome at 36 months did not show statistical difference between the groups, although trends favored the PRP group.

The strengths of this study are the long-term follow-up, clear control group, comprehensive symptom and functional documentation, strict inclusion and exclusion criteria, and meticulous demographic data collected. The baseline statistics were similar in both groups, with significant differences only in the sex ratio and mean age. The age difference is probably a unimportant factor, as the mean age in both groups was close to 50 years, which is within the typical age range when epicondylitis usually appears. Moreover, the medical history was similar in the two groups. 

The major limitation of this study is the lack of randomization due to its retrospective nature; therefore, certain biases cannot be avoided. Other limitations include the small sample size, and a possible placebo effect in receiving a new type of treatment. Furthermore, the PRP group included more younger males, which may have affected the outcomes. The mean age was significantly higher in the PRP group, possibly because these patients may have tried more treatment modalities for their condition, thus, they wanted to consider a new promising one. We included both medial and lateral epicondylitis patients because they are essentially overuse diseases that affect different muscle and tendon groups of the forearm, and their treatment is essentially the same. Therefore, differences between them are more semantic than clinically relevant in terms of testing treatment options.

Nevertheless, we have no comprehensive follow-up imaging data of the patients, as the elbow epicondylitis treatments do not require any imaging once the diagnosis is set. Hence, we do not have any imaging data documented for this study because all of the patients with osteoarthritis or other significant findings other than elbow epicondylitis were excluded.

Finally, in the methodology, we lacked comprehensive grip strength data.

There were no relapses in the PRP group, which is important when considering socio-economic factors involved in chronic epicondylitis, such as sick leave length or the need for surgery. Furthermore, these results appear to persist in the long run. Candidates for surgical treatment were patients with persistent debilitating pain for more than 6 to 12 months, and failure with any nonoperative treatment for at least 6 to 12 months. In this study, patients had undergone extensive conservative treatment before having orthopedic consultation; even after that, patients continued with nonoperative treatments, PRP, or physical therapy before surgery was considered. If patients still had debilitating pain during controls despite all of these conservative treatments, then they were offered an option for surgery.

The symptom scores in both groups diminished over time, but the PRP group experienced faster recovery than the PT group. The PT group reached similar symptom scores only after 24 months of follow-up, which is a rather long time to wait. Interestingly, VAS, PRTEE pain, and PRTEE total scores were still significantly lower in the PRP group than in the PT group at 24 months, while at 36 months there were no significant differences between the groups, except for the need for surgery. This may be due to the self-limiting nature of the disease. We documented one adverse effect from a PRP injection, as the patient experienced prolonged pain and local swelling of the injection site; it spontaneously resolved after five days. Thus, PRP treatment can be considered a very safe option.

Although the present findings are interesting, the main limitation of this study pertains to its retrospective nature. Furthermore, the nonrandomized design and indications for the type of injections may introduce confounding factors, particularly of unmeasured data. Thirdly, demographic differences between the groups for age and sex ratio may have played a role in final outcomes.

## 5. Conclusions

Patients who underwent PRP injections for epicondylitis resulted in better pain and functional outcomes compared to physiotherapy, and this improvement lasted at least 24 months. The results of this study support the use of PRP injections in chronic elbow epicondylitis when other conservative treatments have failed. PRP injections resolved the prolonged symptoms rapidly and consistently, and, at the same time, patients avoided surgery. When considering socio-economic factors, avoided risks, and gained benefits, it would be reasonable to suggest using PRP injections for chronic elbow epicondylitis. Surgery should be treated with caution, and only reserved as the last treatment option. Further studies should focus on PRP as a first-line treatment for epicondylitis, and determining the possible recurrence rate in long-term follow-up.

## Figures and Tables

**Figure 1 jcm-12-00102-f001:**
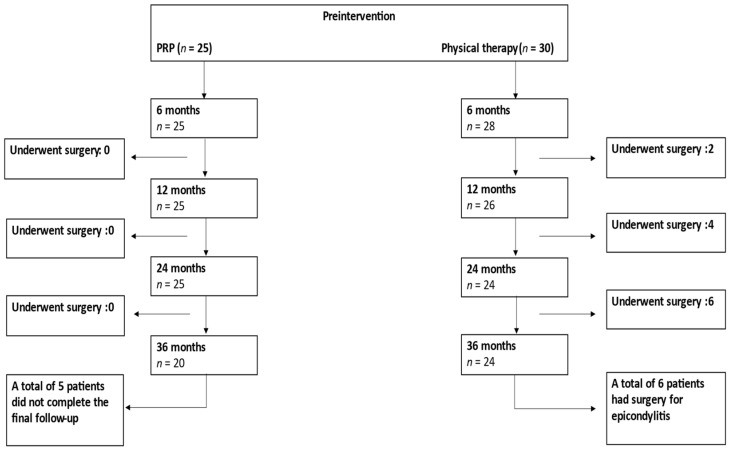
Patient flow during follow-up. PRP = Platelet-Rich-Plasma.

**Figure 2 jcm-12-00102-f002:**
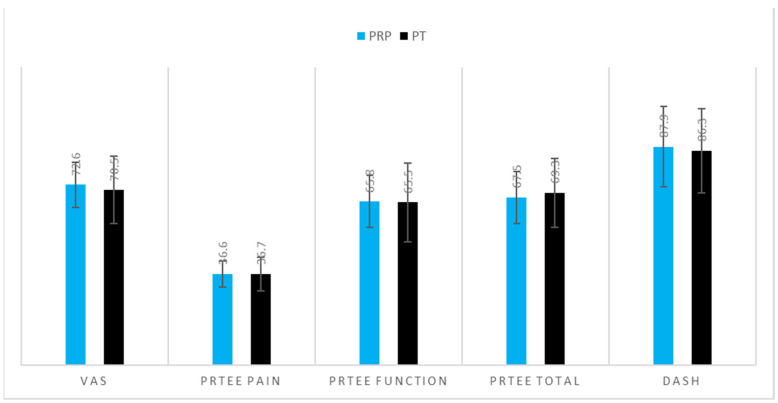
Comparison of pre-interventional parameters in the two groups of patients with epicondylitis. Values are expressed in terms of mean values with ±1 S.D. No significant differences were detected.

**Figure 3 jcm-12-00102-f003:**
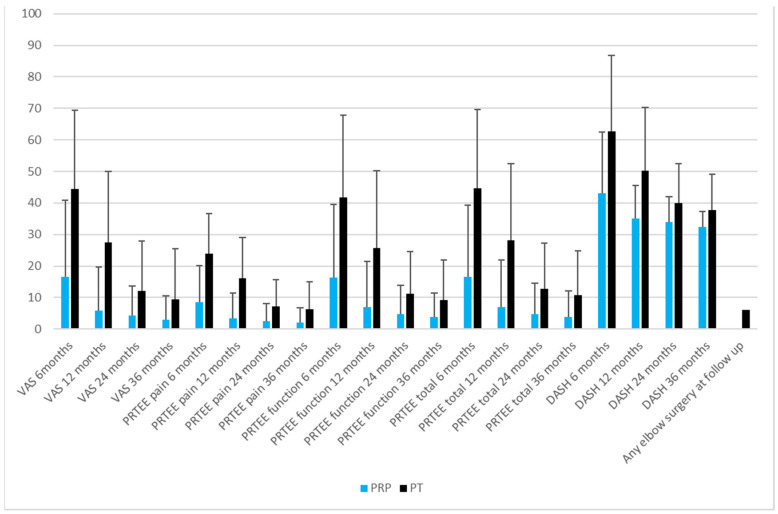
Comparison of post-interventional parameters and need for surgery in the two groups of patients with epicondylitis at 6, 12, 24 and 36-month follow-up. Values are expressed in terms of mean values with ±1 S.D.

**Table 1 jcm-12-00102-t001:** Demographics of the patients with epicondylitis at the time of study. Age, BMI, and follow-up are expressed in terms of mean values with ±1 S.D. An asterisk (*) signifies a statistically significant difference.

	PRP Group (*n* = 25)	Physical Therapy Group (*n* = 30)	*p*-Value
Age (mean ± SD)	53.6 ± 8.4	48.4 ± 9.9	0.045 *
Sex ratio (F:M)	11:14	23:7	0.013 *
Mean BMI (kg/m^2^)	26.7 ± 3.3	28.7 ± 5.1	0.101
Smokers	4 (16.0%)	6 (20.0%)	0.741
Any comorbidity	7 (28.0%)	5 (16.7%)	0.311
Obesity (BMI ≥ 30 kg/m^2^)	3 (12.0%)	10 (33.3%)	0.110
Diabetics	2 (8.0%)	0 (0.0%)	0.202
Smokers	4 (11.4%)	7 (17.5%)	0.528
Rheumatic disease	0 (0.0%)	1 (3.3%)	1.000
Hypertension	7 (28.0%)	4 (13.3%)	0.198
Lipid disease	3 (12.0%)	2 (6.7%)	0.650
Cardiac disease	0 (0.0%)	0 (0.0%)	1.000
Medial/Lateral ratio	2:23	5:25	0.436
Bilateral disease	1 (4.0%)	3 (10.0%)	0.617
Follow-up	40.1 ± 6.5	36.8 ± 15.5	0.329

BMI = Body Mass Index. F:M = Females:Males Ratio.

## Data Availability

The data presented in this study are available on request from the corresponding author.

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
