# Peer review of "Platelet-Rich Plasma Injections Decrease the Need for Any Surgical Procedure for Chronic Epicondylitis versus Conservative Treatment—A Comparative Study with Long-Term Follow-Up"

_jcm, 2022, doi:10.3390/jcm12010102_

Round 1

Reviewer 1 Report

The authors of the study aimed to compare patients with chronic lateral or medial elbow epicondylitis who received a single PRP injection with patients who continued with physical therapy and NSAID.

The study demonstrates that a single injection of PRP can significantly reduce the need for surgery and improve pain and function scores in patients affected by elbow epicondylitis.

The major limitation of the study is its lack of randomization and choosing the need for surgery as the primary outcome measure. The authors don’t explain how patients were divided into PRP and physiotherapy groups. This resulted in non-homogenous groups – the PRP group included more younger males. The primary end point of the study is surgery. Again, the authors don’t explain how the decision to proceed with surgery was made; was is the surgeon who is not blinded to the treatment pushing patients to surgery because he has no other treatment options? Was it the patient’s hope that the novel treatment is better? Are there different psychological considerations for the young male patient in the physiotherapy group who prefers surgery compared to older female patients in the PRP group?

It is better if the authors concentrate on objective findings – PRP injection resulted in better pain and functional outcomes compared to physiotherapy, and this improvement lasted at least 24 months. This finding is consistent with previous studies demonstrating beneficial effect of PRP treatment. Interestingly, final outcome at 36 months did not show statistical difference between the groups.

Other minor remarks:

1.       I am not sure if medial and lateral epicondylitis should be included together as one disease. If the authors choose to combine them, please state how many medial and lateral epicondylitis were in each group.

2.       Authors should elaborate about the PRP used – is it leucocyte rich or poor, was the injection done under ultrasound guidance, etc.

Author Response

We wish to thank each of the Reviewers for their thoughtful and insightful comments on our manuscript entitled: “Platelet-rich plasma injections decrease the need for any surgical procedure for chronic epicondylitis versus conservative treatment – a comparative study with long-term follow-up”.

We have done our best to address each of the comments in our revised manuscript and feel that the changes that have been made have resulted in an even better manuscript that will be of use to all physicians who are involved with epicondylitis treatment and platelet-rich plasma therapy.

A point-by-point response to the Reviewers’ comments follows.  Changes are reported in red.

Comment: 1.    The authors of the study aimed to compare patients with chronic lateral or medial elbow epicondylitis who received a single PRP injection with patients who continued with physical therapy and NSAID.

The study demonstrates that a single injection of PRP can significantly reduce the need for surgery and improve pain and function scores in patients affected by elbow epicondylitis.

Response: We are very grateful for your comments. 

Comment: 2.    The major limitation of the study is its lack of randomization and choosing the need for surgery as the primary outcome measure.

Response: Thank you for these comments. The lack of randomization has been discussed as major limitation of this study (Please see page 7). We have now changed the primary outcome measure as you recommended including pain and functional outcomes. (Please see pages 1 and 3)

Comment: 3.    The authors don’t explain how patients were divided into PRP and physiotherapy groups. This resulted in non-homogenous groups – the PRP group included more younger males.

Response: Thank you. We have now stated it into the Methods section how patients were divided. Patients were referred for orthopedic consultation by primary healthcare and they were given option to try physical therapy or PRP injection therapy to treat their condition. We also have discussed the fact that PRP group included more younger males as limitation (Please see pages 3 and 7)

Comment: 4. Again, the authors don’t explain how the decision to proceed with surgery was made; was is the surgeon who is not blinded to the treatment pushing patients to surgery because he has no other treatment options? Was it the patient’s hope that the novel treatment is better? Are there different psychological considerations for the young male patient in the physiotherapy group who prefers surgery compared to older female patients in the PRP group?

Response: Thank you for your excellent question. We have now stated it into the Method section how the decision to proceed with surgery was made. Indication for surgical intervention for elbow epicondylitis was at surgeon’s discretion when failure of prolonged nonoperative treatment (in this study physical therapy and pain medication) persisted over 6 months. This issue has also discussed. Similarly, we have discussed the age differences. (Please see pages 3 and 9)

Comment: 5. It is better if the authors concentrate on objective findings – PRP injection resulted in better pain and functional outcomes compared to physiotherapy, and this improvement lasted at least 24 months. This finding is consistent with previous studies demonstrating beneficial effect of PRP treatment. Interestingly, final outcome at 36 months did not show statistical difference between the groups.

Response: Thank you. We have now stated it into the Abstract and Results. (Please see pages 1, 4 and 8)

Comment: 6. I am not sure if medial and lateral epicondylitis should be included together as one disease. If the authors choose to combine them, please state how many medial and lateral epicondylitis were in each group.

Response: Thank you for your suggestion. We have now stated it into Table 1, Results and Discussion section. We included both medial and lateral epicondylitis patients because they are essentially overuse diseases that affect different muscle and tendon groups of the forearm and their treatment is essentially the same. Therefore, differences between them are more semantic than clinically relevant in terms of testing treatment options.  (Please see pages 4, 5, 7 and Table 1).

Comment: 7. Authors should elaborate about the PRP used – is it leucocyte rich or poor, was the injection done under ultrasound guidance, etc.

Response: Thank you, it was leucocyte rich and injected without ultrasound guidance. We have now stated it into the Method section, as well as how it was injected. (Please see page 3)

Reviewer 2 Report

I am very interested in this field, and you did hard work for this clinical research. But there are some problems, please see below.

I don't see how you could have divided them into two groups. Randomized? or some reason? This point is the big problem. Please add a note about this.

For me, Fig4 is hard to see.

Have you done any MRI evaluations, etc.? 

Author Response

We wish to thank each of the Reviewers for their thoughtful and insightful comments on our manuscript entitled: “Platelet-rich plasma injections decrease the need for any surgical procedure for chronic epicondylitis versus conservative treatment – a comparative study with long-term follow-up”.

We have done our best to address each of the comments in our revised manuscript and feel that the changes that have been made have resulted in an even better manuscript that will be of use to all physicians who are involved with epicondylitis treatment and platelet-rich plasma therapy.

A point-by-point response to the Reviewers’ comments follows.  Changes are reported in red.

Comment: 1) I am very interested in this field, and you did hard work for this clinical research. But there are some problems, please see below. I don't see how you could have divided them into two groups. Randomized? or some reason? This point is the big problem. Please add a note about this.

Response: Thank you for your comments. This is a retrospective study and the lack of randomization is the major limitation. It has been stated into the Discussion section. Please see page 7.

Comment: 2) For me, Fig4 is hard to see.

Response:  Thank you for this point. We have now enlarged it. (Please see Figure 3).

Comment: 3) Have you done any MRI evaluations, etc.?

Response: Thank you for highlighting this. The patients were screened for osteoarthritis and bony defects upon consultation to the orthopedist. Some patients may have had an MRI scan or ENMG to rule out other causes, but not all. We have no comprehensive follow-up imaging data of the patients as the elbow epicondylitis treatments do not require any imaging once the diagnosis is set. Therefore, we do not have any imaging data documented for this study because all the patients with osteoarthritis or other significant findings other than elbow epicondylitis were excluded. We have discussed this issue. Please see page 8.

Round 2

Reviewer 1 Report

.

Reviewer 2 Report

It is unfortunate that it is not randomized, but there are comments in the limitation and I think this is fine for this paper.